# Advances on Therapeutic Strategies for Alzheimer’s Disease: From Medicinal Plant to Nanotechnology

**DOI:** 10.3390/molecules27154839

**Published:** 2022-07-28

**Authors:** Nasser A. Hassan, Asma K. Alshamari, Allam A. Hassan, Mohamed G. Elharrif, Abdullah M. Alhajri, Mohammed Sattam, Reham R. Khattab

**Affiliations:** 1Department of Pharmaceutical Sciences, College of Pharmacy, Shaqra University, Shaqra 11961, Saudi Arabia; abdullh44.aa@gmail.com (A.M.A.); mohammedsattam8@gmail.com (M.S.); 2Synthetic Unit, Department of Photochemistry, Chemical Industries Research Institute, National Research Centre, Cairo 12622, Egypt; rerybadr@gmail.com; 3Department of Chemistry, College of Science, Ha’il University, Ha’il 81451, Saudi Arabia; ak.alshamari@uoh.edu.sa; 4Department of Chemistry, Faculty of Science, Suez University, Suez 43221, Egypt; allam.hassan@sci.suezuni.edu.eg; 5Department of Chemistry, College of Science, Shaqra University, Shaqra 11961, Saudi Arabia; 6Department of Basic Medical Sciences, College of Medicine, Shaqra University, Shaqra 11961, Saudi Arabia; al_harrif@yahoo.com

**Keywords:** Alzheimer’s disease, gene therapy, pathophysiology, nanomedicines, phytopharmaceuticals, neurofibrillary tangles, nutraceuticals

## Abstract

Alzheimer’s disease (AD) is a chronic dysfunction of neurons in the brain leading to dementia. It is characterized by gradual mental failure, abnormal cognitive functioning, personality changes, diminished verbal fluency, and speech impairment. It is caused by neuronal injury in the cerebral cortex and hippocampal area of the brain. The number of individuals with AD is growing at a quick rate. The pathology behind AD is the progress of intraneuronal fibrillary tangles, accumulation of amyloid plaque, loss of cholinergic neurons, and decrease in choline acetyltransferase. Unfortunately, AD cannot be cured, but its progression can be delayed. Various FDA-approved inhibitors of cholinesterase enzyme such as rivastigmine, galantamine, donepezil, and NDMA receptor inhibitors (memantine), are available to manage the symptoms of AD. An exhaustive literature survey was carried out using SciFinder’s reports from Alzheimer’s Association, PubMed, and Clinical Trials.org. The literature was explored thoroughly to obtain information on the various available strategies to prevent AD. In the context of the present scenario, several strategies are being tried including the clinical trials for the treatment of AD. We have discussed pathophysiology, various targets, FDA-approved drugs, and various drugs in clinical trials against AD. The goal of this study is to shed light on current developments and treatment options, utilizing phytopharmaceuticals, nanomedicines, nutraceuticals, and gene therapy.

## 1. Introduction

AD is a gradual age-related, irreversible neurodegenerative disease indicated by behavioral disturbances and cognitive deficits. Various pathogenic routes are engaged in the progression and development of AD, including the formation of plaque, oxidative stress, cholinergic deficit, and cholinergic deficit [1]. Amyloidosis is a large group of pathologic conditions in which a particular type of protein, called amyloid, is abnormally deposited in various tissues or organs [2]. In addition, the creation of neurofibrillary tangles and senile plaques (SP) remains the extremely significant neuropathological feature of AD. Amyloid-beta (Aβ) peptide’s major component is SP which is enclosed by dystrophic neurites and microglia. Due to the altered proteolytic processing of amyloid precursor protein (APP) by β-secretase and γ-secretase, Aβ becomes accumulated.

In the brain of patients with AD, there is an excessive level of Aβ self-aggregation into Aβ oligomers observed. These oligomers are extremely toxic and trigger synaptic dysfunction because of inflammation and oxidative stress [3]. The tiny oligomers create protofibrils, which are less structured than mature fibrils, due to noncovalent interactions. Furthermore, the loss of cholinergic neurons in the cerebral cortex, amygdala, hippocampus, and subcortical nuclei hastens the cognitive decline. [4].

Narrowing of gyri, cortical atrophy because of cholinergic neuron death, broadening of sulci, general shrinking of tissue of the brain, and swelling of ventricles that carry cerebrospinal fluid (CSF) are some of the anatomical abnormalities observed in the brains of AD patients. As AD progresses across the cerebral cortex, judgment deteriorates, which can lead to emotional outbursts and impairment of language [5,6]. Figure 1 depicts the physiological structure of the brain and neurons in Figure 1a, a healthy brain, and Figure 1b, an AD brain.

Typically, a decade or so passes before the illness has taken its course and patients die in a completely helpless state. The long duration of AD and its attack on the fragile structures that harbor the very essence of who we are place an enormous emotional and financial burden on patients, their families, and society. There are still no effective treatments to prevent, halt, or reverse Alzheimer’s disease, but research advances over the past decades could change this gloomy picture. Genetic studies demonstrate that the disease has multiple causes. Interdisciplinary approaches have revealed some of its molecular mechanisms. Progress in chemistry and systems biology is beginning to provide useful biomarkers, and the emergence of personalized medicine is poised to transform pharmaceutical development and clinical trials.

Here we focus on current AD therapeutic strategies which comprise of mechanism-based approaches, pathophysiology, various targets, FDA-approved drugs, and various drugs in clinical trials against AD. 

The FDA-approved drugs that are currently utilized to treat AD are cholinesterase inhibitors, N-methyl D-aspartate (NMDA) receptor antagonists, and combination therapy which are now utilized to treat AD. The depletion of the neurotransmitter is responsible for AD symptoms, which is why cholinesterase inhibitors (such as rivastigmine, donepezil, tacrine (discontinued from 2013), and galantamine) are used to increase ACh levels. These inhibitors restrict the decrease in ACh levels in the brain [7]. Figure 2 shows the chemical structure of FDA-approved drugs.

The NMDA receptor ideally acts by allowing entry of calcium ions to the cell for neurotransmission. However, in AD, the receptor exhibits excessive activity, resulting in excess of Ca^2+^, excitotoxicity, and cell death [8]. Memantine is a non-competitive antagonist that lowers the NMDA-high receptor’s activity by interacting with the open state of the receptor [9].

In the moderate to the severe category of AD patients, a combination of memantine and donepezil has proved to be beneficial in treating symptoms such as cognitive judgment, language, and behavioral issues. The results were significantly better than a placebo [10]. In patients with mild to severe illness, however, the combination proved ineffective [11]. Unfortunately, the currently approved medications only provide short relief from the symptoms of AD; thus, researchers are working to find and develop new treatments for AD.

To effectively halt the progression of chronic progressive illnesses, two or more medicines are frequently required. Probably, trials using “dirty pharmaceuticals” having an impact on various receptor targets such as anti-amyloid and anti-tau effects, neurotransmitter modulation, anti-neuroinflammatory and neuroprotective benefits, and cognitive improvement may be appropriate. Clinical trials are being conducted using a variety of anti-Aβ monoclonal antibodies and other medicines, including nutraceuticals [12,13]. A few examples of drugs under clinical trials are stated in Table 1.

The paper focuses on current developments and treatment options, utilizing phytopharmaceuticals, nanomedicines, nutraceuticals, and gene therapy.

### 1.1. Phases of Alzheimer’s Disease

The different phases of AD can be divided into different categories: (1) the preclinical stage, which is known as the pre-symptomatic stage, lasts for many years or longer. Loss of memory and early degenerative changes in the cortex and hippocampus characterize the major signs of this stage with no functional abnormality in routine activities. Additionally, there are no signs and symptoms of AD that occurred at this stage [14,15,16]. (2) The mild or early stage of AD is characterized by loss of focus and memory, disorientation of location and time, mood swings, and depression [17,18]. (3) In the moderate stage of AD, there is an expansion of the disease to a portion of the cerebral cortex that leads to loss of memory, difficulty acknowledging relatives and friends, impulse control loss, and difficulties in speaking, reading, and writing [19,20]. (4) In severe stage AD, there is the expansion of disease to the whole cortex area that leads to gradually cognitive and functional impairment resulting in an inability to identify the family members and difficulties in urination and swallowing, ultimately causing the death of patients because of these difficulties [21,22,23,24,25]. Different phases of AD are shown in Figure 3.

### 1.2. Cause and Risk Factors of Alzheimer’s Disease

The progression of AD could be caused by several causes, although the specific cause is not known and could be of several origins. In the past few years, genetic and lifestyle variables have been discovered as factors impacting the likelihood of growing AD.

However, age is the most important risk factor, with a high probability of having AD as one grows older. The prevalence of AD is increasing rapidly as global life expectancy rises. However, it has been noted that the risk can be lowered by taking into account the elements that affect the progression of AD (e.g., by keeping a healthy lifestyle and diet). Factors influencing the risk of AD development are shown in Figure 4.

#### 1.2.1. Health and Lifestyle Factors

High levels of cholesterol, high BP (blood pressure), obesity, and diabetes, all of which are associated with cardiovascular illness, can raise the chance of acquiring AD. Though the exact link between cardiovascular disease and dementia is unknown, it is hypothesized that heredity and shared risk factors cause dementia to develop [26].

To avoid these risk factors, people should adopt a healthy lifestyle (e.g., quit smoking, drink less, eat well, and exercise regularly).

#### 1.2.2. Environmental Factors

Low-level chronic exposure to heavy metals and pesticides, such as aluminum and lead, has also been linked to a gradual deterioration in mental functioning and an increased risk of neurodegenerative illnesses. There is emerging evidence that fungi and pollutants are associated with AD development. They are extremely oxidative and relate to unfolded protein accumulation, mitochondrial dysfunction, apoptotic signaling, and calcium homeostasis, all known hallmarks of AD [27].

#### 1.2.3. Genetic Factors

Familial AD (FAD) is linked to an earlier commencement of symptoms and is responsible for about 2–3% of all cases of AD. FAD has been linked to several rare autosomal dominant point mutations. On chromosome 21, mutations in the APP gene can increase the incidence of Aβ (42 amino acids), which aggregates more than the shorter form (40 amino acids) [28].

Presenilin 1 (PSEN1) and presenilin 2 (PSEN2) genes, which determine the two subunits of the enzyme secretase, have also been discovered to be prone to mutations. This enzyme cleaves to APP and converts it to Aβ peptides. Apolipoprotein E4 homozygosity has been identified as a significant risk allele for AD, with an estimated 8-fold increase in risk [29].

## 2. Epidemiology

The annual incidence rate of AD appears to be decreasing, which has been linked to advancements in Alzheimer’s risk factors over the twentieth century [30]. Despite this potentially decreased incidence rate, the number of individuals with AD is anticipated to rise due to an enhancement in the number of adults aged 65 and higher, who are at an elevated risk of AD [31,32,33,34,35,36,37,38]. Incidence rate is a measure of the frequency with which the disease occurs over a specific period. AD affects about one in every nine people aged 65 and up (10.7 percent). According to the Alzheimer’s Association survey, this disease is expected to increase up to 12.7 million in the population of 65 years age and more [39,40].

As per the reports, the mortality rate due to AD has increased up to double in the past two years [41]. A mortality rate is the number of deaths due to a disease divided by the total population. The parameters such as incidence rate and mortality rate helps in calculating average duration of disease. Seniors with AD are twice as likely to die before the age of 80 as those without the disease. As per the available epidemiological data, as of today, approximately 24.3 million people are suffering from AD which is expected to increase by about 4.6 million new cases of AD every year 1.e. in every 7 s: one new case AD. Every 20 years, the number of persons affected will double, reaching 81.1 million by 2040 [41,42,43,44]. Around 200,000 people with AD are under the age of 65, although the vast majority (81%) are beyond the age of 75.

## 3. Pathophysiology

The hippocampus, cortical association areas of the temporal, frontal, parietal cortices, entorhinal cortex, and amygdala show loss of neurons and/or pathology. Trans-entorhinal tangles spread to the entorhinal cortex, CA1 region of the hippocampus, and finally, the cortical association areas, which are impacted in the frontal, parietal, and temporal lobes. More than amyloid plaques, tangle formation is connected to dementia severity [45].

Multiple factors are responsible for the progression of AD. The most popular factors are amyloid plaques and hyperphosphorylated tau. Senile plaques are formed because of extracellular amyloid deposition. Hyperphosphorylated tau causes microtubule disintegration and harms the cytoskeleton and signal transduction systems in neuronal cells. Other responsible factors that contribute to the progression of disease include cholinergic insufficiency, oxidative stress, dysfunction of mitochondria, neuroinflammation, and autophagy dysfunction. The presence of macrophages and monocytes in the cerebral cortex leads to the activation of microglial cells in the parenchyma [46]. Various hypotheses for AD are listed in detail below (Figure 5).

### 3.1. Hyperphosphorylated Tau Protein and Aβ Hypothesis

SP production by Aβ deposition is one of the pathogenic characteristics of AD. Aβ is a soluble micro peptide produced by APP breakdown by γ-secretase. It can lead to dangerous oligomers such as protofibrils, fibrils, or plaques depending on the extent of oligomerization. The cause of Aβ is unknown, but Aβ’s stability, concentration, and sequence are all critical factors for AD development. Tau, on the other hand, is a protein that is linked to microtubules and promotes their stability and polymerization; these functions are controlled by tau phosphorylation. The defective metabolism of APP and improper clearance of amyloid cause a cascade of events, including hyperphosphorylated tau-mediated disassembly of microtubules and synaptic failure, which leads to AD [47].

### 3.2. Oxidative Stress Conditions

ROS and RNS (reactive oxygen and nitrogen species) are formed in a range of normal and pathological processes in humans. They provide a dual purpose in that they have both positive and negative impacts on cellular signaling pathways, as well as the ability to destroy cellular structures (including DNA, cell membrane and lipid, protein).

The brain is especially susceptible to oxidative stress due to its high oxygen consumption, which uses 20% more oxygen than other respiratory tissues of mitochondria. The brain’s basic functional unit, the neuron, contains a range of polyunsaturated fatty acids. Moreover, glutathione depletion in neurons induces oxidative stress damage [48].

### 3.3. Unbalance of Metal Ions

Metal dyshomeostasis has a great contribution to the pathophysiology and progression of several disorders, including neurodegenerative diseases. In addition to the ionosphere and metal chelators, several other chemicals are now being studied in clinical trials.

Drugs that target transition metal homeostasis are not limited to metal-binding molecules. The redox transition metal, particularly copper (Cu), iron (Fe), and other trace metals, alters the normal physiology of brain, and their levels in the brain are higher in AD [49]. Copper, manganese, aluminum, and zinc have all been linked to a variety of neurological disorders [49].

### 3.4. Cholinergic Theory

The APOE genotype influences the efficacy of AChEIs in AD patients. The AchEI drugs are vital in treating AD, and the APOE genotype is the most important risk factor. Since the “Cholinergic Theory” of AD in 1976, it has been recognized that cholinergic neurons are not the disease’s main target.

In mild to moderate Alzheimer’s, cholinergic receptor binding is reduced in various brain regions, causing neuropsychiatric symptoms. This decrease in receptor binding may be linked to reduced processing speed in elderly adults [50]. For almost 20 years, cholinesterase inhibitors (ChEIs) and donepezil have been used to treat symptoms of AD [51]. By binding to AchE and inhibiting reversible Ach hydrolysis, donepezil raises the concentration of ACh at synapses. The treatment is widely accepted, and mild and transient cholinergic side effects only have a temporary impact on the digestive and nervous systems [50].

### 3.5. Amyloid Cascade Theory

The amyloid cascade hypothesis postulates that neurodegeneration in AD is caused by the abnormal accumulation of amyloid-beta (Aβ) plaques in various areas of the brain. In addition to this, in silico studies extended the views of Aβ peptide recognition and its cleavage by Aβ degrading enzymes. Therefore, computational studies will offer a wealth of fairly reliable information regarding the potential of Aβ degrading enzymes as a therapeutic target. The amyloid hypothesis has continued to gain support over the last two decades, particularly from genetic studies.

## 4. Targets of Alzheimer’s Disease

Following a detailed understanding of disease through its pathophysiology, the various targets for AD are discussed in detail below.

### 4.1. Acetylcholinesterase

AChE is an enzyme that is responsible for neurotransmission at cholinergic synaptic connections to be terminated. Acetylcholine hydrolysis is accelerated by this enzyme. AChE reduces cholinergic activity, which has been linked to the degree of cognitive impairment and decreased levels of the neurotransmitter ACh. These drugs compensate for the death of cholinergic neurons and offer symptomatic relief by inhibiting acetylcholine (ACh) turnover and restoring synaptic levels of this neurotransmitter [52,53].

ACh binds to AChE and is degraded to choline and acetate, effectively terminating its function as a neurotransmitter. Additionally, the enzyme has increased catalytic activity at low concentrations of ACh. Amyloidogenic factor AChE appears to modulate amyloid fibrils’ toxicity in vitro and in vivo. The incorporation of AChE into Alzheimer’s amyloid aggregates results in the formation of stable complexes that change the biochemical and pharmacological properties of the enzyme, and cause an increase in the neurotoxicity of the amyloid-β fibrils, suggesting that AChE could play a pathogenic role in AD by influencing the process leading to amyloid toxicity and the appearance of AD [54]. AChE inhibitors stop ACh from being degraded after it leaves cholinergic terminals. In addition to symptomatic relief, AChE inhibitors appear to slow the progression of the disease. The peripheral anionic site (PAS) of AChE, located at the entrance to the active site gorge, inhibits substrate binding. Among the several drugs used to treat AD, donepezil, rivastigmine, and galantamine all work by inhibiting AChE activity [55,56].

### 4.2. NMDA Receptor

NMDA receptors, on the other hand, have gained a lot of interest recently due to their role in excitotoxicity and various forms of brain plasticity. It is very important to establish clinically useful NMDA receptor antagonists that inhibit excitotoxic activation of NMDA receptors without affecting the function of the NMDA receptor. This is generally required for normal synaptic transmission and plasticity. According to recent research, neurosteroids are NMDA receptor inhibitors with favorable qualities. These medications result in voltage-independent NMDA blockage and lower the risk of AD [57,58].

Memantine is a non-competitive glutamatergic NMDA receptor antagonist that is frequently used to treat AD. Memantine has been shown to lower Aβ levels in both neuronal cultures and the brains of animal models of AD. Memantine affects APP’s endocytosis route, which is necessary for cleavage by β-secretase. As a result, the amount of APP in the brain decreases, lowering the risk of AD [59,60].

### 4.3. Glycogen Synthase Kinase (GSK3)

GSK3 is a proline-directed serine/threonine kinase present throughout the body that participates in several physiological activities such as transcription of genes and metabolism of glycogen. The formation of neurofibrillary tangles by hyperphosphorylation of tau is a critical event in the pathophysiology of AD. MiR-124-3p belongs to the microRNA (miRNA) family and was markedly decreased in AD; however, the functions of miR-124-3p in the pathogenesis of AD remain unknown [61]. There is a decrease in miR-124-3p expression in N2a/APP695 cells, as well as an increase in aberrant tau hyperphosphorylation. The overexpression of GSK3beta increases the expression of Caveolin-1, PI3K, Akt-Ser473, and GSK-3 in N2a/APP695swe cells. Mutations in Caveolin-1-PI3K/Akt/GSK3 prevent tau hyperphosphorylation in AD. Thus, miR-124-3p appears to have neuroprotective properties in AD, leading to new treatment targets and concepts [61,62,63].

### 4.4. BACE1

β-site APP cleaving enzyme 1 (BACE1) is the β-secretase enzyme that produces Aβ peptides which, in the neuronal cells, accumulate as fibrillary plaque. BACE1 was initially cloned and characterized in 1999. It is required for the generation of all monomeric forms of amyloid-β (Aβ), including Aβ_42_, which aggregates into bioactive conformational species and likely initiates toxicity in AD. BACE1 concentrations and rates of activity are increased in AD brains and body fluids, thereby supporting the hypothesis that BACE1 plays a critical role in AD pathophysiology. Therefore, BACE1 is a prime drug target for slowing down Aβ production in early AD. Besides the amyloidogenic pathway, BACE1 has other substrates that may be important for synaptic plasticity and synaptic homeostasis [64]. These induce degeneration of the brain, neuronal malfunction, neuroinflammation and death, which are the primary causes of pathogenesis in AD. Mutations in three genes (APP, PSEN1, and PSEN2) cause FAD to manifest early by enhancing Aβ peptide formation. It is produced via the amyloidogenic pathway’s cleavage of the APP by β- and γ-secretase [65,66].

The non-amyloidogenic process is the normal pathway in which the cleavage of APP due to α- and γ-secretase forms soluble peptide fragments. In the amyloidogenic process, the cleavage of APP due to β-secretase at the Asp+1 residue of the Aβ sequence forms the peptide’s N-terminus. C99, a membrane-bound carboxyl-terminal fragment (CTF), remains attached to the structure. To lower the Aβ levels, BACE1 activity must be reduced [67,68].

## 5. Evolution of Treatment Strategies

The evolution of treatment strategies of disease starting from medicinal plants to nutraceuticals, then pharmacological approaches followed by treatment through nanotechnology and then gene therapy is discussed in detail below.

### 5.1. Medicinal Plants Used in the Treatment of AD

Medicinal plants include phytocompounds that can be extracted and used in other scientific studies. Pharmaceutical companies use a variety of plant secondary metabolites. Medicinal plants have recently acquired popularity due to their fewer side effects compared to manufactured drugs and the need to address the growing human population’s medical needs. However, broad geographical distribution, environmental changes, cultural practices, labor costs, selection of superior plant stock, and over-exploitation by pharmaceutical businesses make a stable supply of source material challenging.

Anti-inflammatory herbs such as ginseng, turmeric, licorice and sweet flag, and many others, reduce inflammation of the brain tissue in AD. The neurochemical Ach plays a vital role in cognition and reasoning. The brains of individuals with mild-to-moderate AD have exceptionally lower Ach concentrations [69]. Various herbs with their family, plant extract, phytoconstituents, and probable mechanism of action are indicated in Table 2 given below.

### 5.2. Nutraceuticals

The term “nutraceutical” combines the two words: “nutrient” which is a nourishing food component and “pharmaceutical” which is a medical drug. These are any products derived from food sources with extra health benefits in addition to the basic nutritional value found in foods. Several chemical substances are belonging to classes of natural dietary origin display protective properties against some age-related diseases, including neurodegenerative ones, particularly AD. Nowadays, there has been a lot of focus on developing food-based therapeutic approaches to minimize the progression of the disease and alleviate symptoms. These food-based therapeutic formulations are rich in bioactive chemicals that have special health benefits in addition to providing basic nourishment.

In the future, a combination of modern bioactive chemical understanding and traditional food formulation expertise will lead to the development of disease-specific treatment regimens to prevent and cure a variety of chronic diseases. Ayurveda, an herbal-based traditional system of remedies and treatment procedures that originated in India and the Indian subcontinent, provides various unique strategies for preventing, delaying, and curing different types of mental disorders, including AD.

Various formulations promise to improve memory retention and cognitive functions along with slowing down age-related mental diseases. Garlic extract (Allium sativum), turmeric (Curcuma longa), and other local plants provide some of the essential bioactive compounds that could be used as nutraceutical components. The efficacy and efficiency of these plants, which have been studied for over 2000 years, demonstrate the depth of knowledge of the Indian traditional medicine system.

Nonetheless, the necessity for extensive pharmacological and toxicological studies has attracted the attention of researchers all over the world in recent years, and investigations are underway to verify the purported pharmacological and therapeutic outcomes. The present contribution will attend to the most popular and effective nutraceuticals with proposed brief mechanisms entailing antioxidant, anti-inflammatory, autophagy regulation, mitochondrial homeostasis, and more. Therefore, even though the effectiveness of nutraceuticals cannot be dismissed, it is essential to do further high-quality randomized clinical trials.

Nutraceuticals and a diet rich in nutraceuticals play a key role in preventing the onset of many disorders and limiting the issues that arise because of these diseases. Galantamine is the first nutraceutical to be licensed by the FDA for reversible AChE inhibitory activity and can be derived from *Leucojum atrium*, *Narcissus tazetta*, *Lycoris radiate*, and *Galanthus nivalis*. It has revived interest in plant therapies for AD [108,109,110].

#### 5.2.1. Aged Garlic Extract (AGE)

Nutraceuticals have also shown anti-amyloidogenic properties. AGE suppresses the development of ROS that are found to be involved in the apoptotic mechanism of Aβ -mediated neurotoxicity [111].

#### 5.2.2. Alpha-Lipoic Acid

The symptoms of AD such as a discrepancy between free radicals and antioxidants, as well as a reduction in neuronal energy, were found to be lower when people with AD took 600 mg of alpha-lipoic acid once a day. This may be utilized as a supplement in “neuroprotective” treatment [91].

#### 5.2.3. Caffeine and Chlorogenic Acids

These compounds derived from *Arabian coffee* may help to reduce the symptoms of AD. This effect is based on the neuronal factor nicotinamide mononucleotide adenylyl transferase 2, which has been demonstrated preclinically in a mouse model of tauopathy (rTg4510) [112,113].

#### 5.2.4. Caprylic Acid

Caprylic acid, a triglyceride-rich component of Axona, has also shown promising results in the treatment of AD. It is made from coconut oil that has been refined. The ketone body, hydroxybutyrate, which is generated during its metabolism in the liver, can be used as a source of strength in the brain. It is an excellent supplement for AD [114].

#### 5.2.5. Cinnamon

Cinnamon extract aids in the suppression of the formation of A oligomers, hence reducing the harmful effect on neuronal PC12 cells. The effect of AD was studied using a fly model, which revealed a shorter lifespan and restored locomotor deficits, as well as a reduction in tetrameric species of A in the brain. As a result, when this natural herbal component is included in the diet of AD, it improves cognitive behavior [115].

#### 5.2.6. Curcumin

Curcumin can be obtained from *Curcuma longa* which contains a hydrophobic polyphenol. It is a spice that can also be used as a topical anti-inflammatory. In addition to its anti-tau function, it is anti-amyloidogenic, down-regulates BACE1 expression, anti-cholinesterase and has anti-secretase action, therefore, this drug is also useful in neurological illnesses such as AD. This product’s distribution using nanoparticles has recently been created for AD [116,117].

#### 5.2.7. Omega-3 Fatty Acid and Fish Liver Oil

Omega-3 fatty acid and fish liver oil have only lately been studied as a supplement for the treatment of AD. There is evidence that they can reverse dementia caused by AD through several brain pathways, including neuroprotection, limiting the expansion and accumulation of Aβ peptide toxin and hindering many Aβ transduction pathways that stimulate kinases and encourage the pathology of neurofibrillary tangles [118].

#### 5.2.8. Grape Seed Extract (GSE)

Polyphenols obtained from grape seeds were evaluated for their capability to decrease Aβ aggregation, which minimizes the Aβ formation and hence protects animals with AD from neurotoxicity. Mice fed a GSE-rich diet had lower levels of Aβ in their brains and serum. In an experimental model involving a mouse, GSE rich in polyphenols prevents Aβ arrangement and lowers inflammation in the brain, indicating that it is beneficial in preventing AD development [119].

#### 5.2.9. Piperine

Piperine, an active alkaloid in *Piper nigrum*, has also shown efficacy against AD. Researchers found that the compound improved the memory impairment in the rat model. It also improved neurodegeneration in the hippocampus [120].

#### 5.2.10. Resveratrol

Trans-3,40,5-trihydroxystilbene is derived from grapevines (*Vitis vinifera*). By causing amyloid breakdown, it improves the chances of removing intracellular amyloid. Resveratrol is said to be in Phase III clinical studies as a food supplement in conjunction with other drugs. By reducing cognitive impairment and oxidative stress, it aids in the repair of brain cell metabolism for proper functioning [121,122].

#### 5.2.11. Quercetin

Although it has been shown to improve memory and learning capacity in patients with AD, its low bioavailability limits its use. It was recently improved as nano encapsulated quercetin and evaluated in the SAMP8 mice animal model to see how effective it is. SAMP8 model is a mouse model of age-related cognition decline with underlying mechanisms in AD [121].

According to estimates, delivering nano encapsulated quercetin orally in zein nanoparticles reduced memory loss and improved cognition in mice [123]. 

Apart from the aforementioned plant products, flavonoids such as hesperidin, naringenin, luteolin, apigenin, and kaemferol, as well as anthocyanidines such as pelargonidine, cyanidine, and malvidin, have been isolated from berry fruits and red wine, and have shown promising effects in boosting neuronal survival and cerebral blood flow with the likely mechanism being the activation of antioxidant enzymes. They might also be able to avoid pathophysiological processes that contribute to the etiology of AD. The various nutraceuticals for the treatment of AD are shown in Figure 6 [124].

### 5.3. Pharmacological Approaches

As an incurable age-related neurodegenerative condition, AD necessitates exact diagnosis, early if possible, and effective etiological treatment, as well as consideration of its pathogenesis [125]. However, current medications are cholinesterase inhibitors and glutamate antagonists that only provide symptomatic alleviation. Treatments based on etiology are currently being tested in clinical studies, along with other complimentary preventative measures such as regular physical activity and a good diet [126]. The symptomatic treatments are described in detail below. 

#### 5.3.1. Cholinergic Inhibitors

According to the cholinergic hypothesis, the decline in ACh production is the root cause of AD. Cholinergic levels can be increased by lowering AChE, a therapy strategy for boosting cognitive and neural cell performance. AChEIs block acetylcholine breakdown in synapses, resulting in constant ACh accumulation and stimulation of cholinergic receptors. To treat AD, tacrine (tetrahydroaminoacridine) was the first cholinesterase inhibitor medicine licensed by the FDA. It was later discovered that certain AChEIs, such as donepezil, rivastigmine, and galantamine, could cure the symptoms of AD [127,128,129,130]. Increasing choline reuptake and, as a result, the formation of Ach at presynaptic terminals may also be beneficial in the treatment of AD [131].

##### Donepezil

Donepezil is an indanonebenzylpiperidine derivative that belongs to the second generation of AChEIs and is extensively employed in the treatment of AD. Taking donepezil causes a higher concentration of ACh at synapses by binding to AchE and inhibiting reversibly Ach hydrolysis [132]. The treatment is widely accepted, and the digestive and nervous systems are only temporarily affected by mild and transient cholinergic side effects. AD symptoms such as improved cognition and behavior are treated with donepezil, a medication that does not affect the disease’s progression [133,134,135].

##### Rivastigmine

To impede the metabolism of ACh, rivastigmine binds to the two active sites of acetylcholinesterase (AChE) and butyrylcholinesterase (BuChE). While only 10% of AChE activity is found in glial cells in the healthy brain, BuChE activity is up to 40–90% in the AD brain, while ACh activity is down, suggesting that BuChE activity may signify mild to severe dementia. As rivastigmine is metabolized at the synapse by AChE and BuChE, it is described as “pseudo-irreversible” [136]. To treat mild to severe AD, rivastigmine is commonly prescribed to patients. Vomiting, dyspepsia, and anorexia are some of the negative effects that might occur when the medicine is taken orally. However, these adverse effects can be handled over time, and as a result, the medicine becomes more acceptable [137,138,139,140,141].

##### Galantamine (GAL)

GAL is used to treat mild to moderate AD. The nicotinic acetylcholine receptor is activated by allosteric binding to the subunit, acting as a competitive inhibitor of AChE. GAL, like other AChE inhibitors, has great efficacy and tolerance [142,143,144,145].

#### 5.3.2. NMDA Antagonists

The NMDA function has an impact on the progression of Alzheimer’s disease. Long-term potentiation (LTP) is necessary for synaptic neurotransmission, plasticity, and memory formation, and is created by NMDA activation, which triggers Ca^2+^ influx and, as a result, enhances signal transduction and gene transcription. The majority of NMDA uncompetitive antagonists tried in clinical trials failed due to ineffectiveness or negative effects. Moderate to severe AD can also be treated with memantine. Due to memantine’s low affinity, it can be swiftly displaced from NMDA by high glutamate concentrations, inhibiting the excitatory receptor without interfering with normal synaptic transmission and preventing a persistent blockage [146,147,148,149]. AChE inhibitors for the treatment of AD along with NDMA antagonist memantine are shown in Figure 7.

### 5.4. Nanotechnology and Treatment of AD

#### 5.4.1. Nanomedicine

For the treatment of AD, several small-sized nanocarriers have been adopted for the safe and efficient delivery of drugs. These nanocarriers with specific drugs belong to nanomedicines. While there is no cure for AD, the function of currently FDA-approved drugs are to treat the symptoms of AD. The current anti-AD drugs can improve the clinical symptoms rather than reverse or prevent the progression of the disease. To deliver these recommended drugs to the affected part of the AD brain, NP-functionalized nanomedicine is considered the most useful and applicable approach. Nanomedicines have a set of unique properties that enable them to deliver the anti-AD drugs at target sites in the brain [150,151]. Nanomedicines have the advantages of reduced dimensions and increased biocompatibility that facilitate easy transport of therapeutic substances into the brain. Small-size (approximately 100–10,000 times smaller than a human cell) nanomedicines can easily interact with the proteins and molecules on the cell surface as well as inside the cell. NP-functionalized nanomedicines have central core structures that ensure the encapsulation or conjugation of drugs and provide the protection and prolonged circulation in the blood. Nanomedicines are also specialized to target cells or even an intracellular compartment like Aβ in cells and thus can deliver the drug at a predetermined dosage directly to the pathological site. Nanomedicines can minimize the dose and frequency and then improve patient compliance. Regardless of some clinical issues, nanomedicines have potential advantages of favorability to the brain, greater stability, biocompatibility, and biodegradability, protection from enzymatic degradation, increased half-life, improved bioavailability, and controlled release over other conventional ways of drug delivery to the brain to cure AD [150]. With a detection tool or affecting nucleation through blocking the intermediates which assemble Aβ, and tau which becomes directly adsorbed on the surface of NPs, the nanomedicines have tailored and transformed both diagnostic and therapeutic approaches for AD. Nanomedicines provide state-of-the-art alternative approaches to overcome the challenges in drug transport across the BBB [150].

The various approaches to nanotechnology are discussed in detail below.

#### 5.4.2. Organic Nanostructures

##### Polymeric-Based NPs

The NPs based on polymeric substances formulated with PEG and antibodies were successfully investigated in transgenic AD mice. It has been observed that NPs in combination with PEG helped in curing memory defects and significantly reduced Aβ-soluble peptides. Thus, it has good applications to treat AD illness [152]. 

To improve the effectiveness of memantine against AD, biodegradable polymeric NPs obtained by the double emulsion technique were loaded with memantine in a reported study. Memantine-loaded NPs reduced Aβ plaques and AD-associated inflammation in AD brain [153]. In addition, vitamin D binding protein in PLGA nanoparticles acted as a therapeutic candidate to treat AD symptoms. It reduced Aβ buildup, neuronal loss, neuroinflammation, and cognitive impairment in transgenic AD mice [154]. NPs loaded with zinc were also useful in decreasing the size of the amyloid plaque and mitigated other dysfunctions in AD mice [155]. The NPs in combination with polysaccharides offered several advantages of being non-toxic, highly stable, hydrophilic, and biodegradable [156,157]. To yield huperzine A, which is an inhibitor of acetylcholinesterase, the mucoadhesive and target PLGA-NPs were adopted with their surfaces modified by lactoferrin-conjugated N-trimethylated chitosan offering good and high efficacy in targeting AD pathologies [158,159,160,161,162].

##### Nanomicellar

A nanomicellar formulation of coenzyme Q10 (UbisolQ10) which is soluble in water transferred to double transgenic AD mice as drinking water. Coenzyme Q10 is most commonly used for conditions that affect the heart, such as heart failure and fluid build-up in the body (congestive heart failure or CHF), chest pain (angina), and high blood pressure. It was observed that this combination improved the long-term memories and also inhibited the level of Aβ plaques circulated [163]. The investigations showed that the combination of micelles with Tween-80 to obtain curcumin micelles increases the efficacy and availability of curcumin to treat AD symptoms [164]. In the recent studies, the impact of PEG ceramide nanomicelles was analyzed on neuronal N2 cells which showed the high efficacy of nanomicelles in mediating tau proteins’ degradation and inducing autophagy in targeted cells. Other investigations displayed that polymeric nanomicelles loaded with curcumin significantly inhibited amyloidogenesis in AD mice which acted as a targeted therapeutic delivery system [165].

##### Dendrimers

Dendrimers are found to be highly effective materials helping in the treatment of AD. The coupling of low-generation dendrimers and lactoferrin gave novel results toward brain-targeted delivery of memantine in AD-induced mice. Studies have shown that dendrimers inhibit the formation of amyloid deposits. They exhibit hydrolytic properties relative to the existing forms of aggregated proteins. After incubation of cells with the dendrimer’s nano-molecules, there were no forms of Aβ resistant to hydrolysis [166].

Recent investigations showed a significant impact on the memory aspects in target mice [167]. To enhance the effectiveness of drugs used for CNS disorders like AD, dendrimers with ethylenediamine are generally employed which increases the solubility and bioavailability of the drug, further enhancing permeations over BBB to target damaged brain parts. The poly(amidoamine) dendrimer and gold NP nanostructures were created for disposable immunological platforms while determining AD biomarkers [168,169]. The structure of dendrimers is essential for the effectiveness against Aβ aggregates. The cytotoxicity of Aβ depends on the degree of aggregation [166].

##### Nanogels

Presently, nanogels are the best drug delivery vehicles having the capacity to hold macromolecules, active molecules, and drugs together, and are also being explored for various kinds of pathologies including AD [170]. In a recent investigation, it was observed that delivering deferoxamine in the form of nanogels along with the use of chitosan and tripolyphosphate through the ionotropic method can serve as a proficient therapy against AD. In a preclinical investigation in mice, it was observed that nanogels serving as a carrier can enhance the nose-to-brain delivery of insulin [171].

#### 5.4.3. Lipid-Based Nanoparticles

Nowadays, lipid-based nanoparticles are extensively contributing to the development of theranostic agents. These nanoparticles are promising theranostic agents for a wide range of applications, including drug delivery, phototherapy, magnetic resonance imaging (MRI), and PET. They end up inside tumors due to the enhanced permeability and retention (EPR) effect and after their entrance into the cancer cells, they are disrupted into lipid subunits. Both the intact and the disrupted state can be exploited for therapeutic purposes [172]. This is followed by modifications in the nano system properties along with the particular drug or ligand addition on their surfaces. Though curcumin possesses the significant potential of stimulating neurogenesis, the applications of this phytochemical in neurodegenerative therapies have various limitations, for example, poorly soluble in water, least bioavailability, and instability. As per the recent data, soft lipid-based nano curcumin was developed which improved its stability and solubility, showing greater inhibition of neuronal loss in AD [173].

##### Solid Lipid NPs

α-Bisabolol is an excellent carrier for solid lipid NPs in the AD brain. Anti-amyloid aggregation is greatly reduced by this formulation P-glycoprotein, and breast cancer resistance protein transporters on brain endothelial cells can be induced by targeting the MC11 ligands, according to a new method. In both in vitro and in vivo experiments, donepezil was added to solid lipid NP formulations, which increased intranasal drug delivery [174,175]. Drug effectiveness was significantly improved when compared to other formulations. The hallmarks of Aβ in AD could be suppressed by these nanostructures as well [176]. 

##### Liposomes

Self-assembling and amphiphilic capabilities of vesicular liposomes at the nanoscale have made them popular as nanocarriers for the delivery of medications to brain tissues. A liposome’s surface can be modified by using different functional proteins, polyethers, and cell-penetrating peptides (CPP) that aid in the transport of drugs over the BBB [177]. Examples of liposomes that can circumvent the opsonization of RES include polyethylene glycol (PEG)-coated liposomes. Liposomes containing glutathione-PEGylated glutathione have also been shown to improve the cellular uptake of the medication across endothelial BBB. Curcumin-loaded liposomes can significantly increase drug delivery to the central nervous system (CNS) by binding to specific receptors on BBB cells. With this liposome carrier system, ApoE2 has been delivered to the brain of an Alzheimer’s patient using a mannose ligand and CPPs-covered surface. Treatment for AD can make use of the high concentration of genes that functionalized liposomes can deliver to the target tissues [178].

##### Niosomes

Niosomes are vesicles composed of non-ionic surfactants, which are biodegradable, relatively nontoxic, more stable and inexpensive, and an alternative to liposomes. Rivastigmine is a renowned acetylcholine esterase inhibitor that can help people with CNS illnesses like AD regain their cognitive abilities. Utilizing the film hydration process, a niosome formulation is created using sorbitan esters and cholesterol. This preparation had outstanding results in terms of increasing medication efficacy in specific brain tissues [179].

Kulkarni et al. recently designed, developed, and evaluated intranasally delivered rivastigmine and n-acetyl cysteine-loaded bifunctional niosomes for applications in combinative treatment of Alzheimer’s disease. In vivo pharmacokinetic and organ biodistribution studies revealed a better drug profile and greater distribution of the niosomes in the brain compared to other organs, thereby indicating a direct nose-to-brain delivery of the niosomes [180,181]. In another case, when treated against amyloid aggregation, the artemisia absinthium-loaded lipid nanocarrier system showed outstanding effects on AD pathology. Since amyloid is a risk factor for AD, this formulation can be used to prevent its formation.

##### Nanoemulsion

In nanoemulsion preparations, anti-AD medication’s potency is exploited in a targeted manner to reach specific brain regions. In vivo and in vitro studies revealed that emulsion has beneficial effects on AD pathology [182]. The naringenin nanoemulsion was created to increase clinical use while improving efficacy. The findings suggest that using a naringenin nanoemulsion to combat Aβ neurotoxicity and amyloidogenesis could be a viable option.

##### Cubosomes

Cubosomes are another lipid-based NP with a wide range of medicinal uses, including drug transport to the brain. The results of donepezil-HCl administration using a cubosomal mucoadhesive in situ nasal gel showed that the prepared gel can be an effective carrier for delivering medicine to the brain’s afflicted areas [183].

##### Amylolipid Nanovesicles

Lipid-based nanocarriers have been developed to get the maximal medication concentration across the BBB, allowing for a more fast and powerful effect on brain cells. It is a self-assembling lipid-modified starch hybrid structure. Curcumin loaded with amylolipid nanovesicles administered intravenously has been shown to have a higher propensity to penetrate the BBB, resulting in considerable effects against the pathology of AD. As a result, the findings show that this carrier system is an effective way to transport drugs to AD brain regions [184].

#### 5.4.4. Metallic Nanoparticles

Using metallic NPs in nanomedicine-based approaches to the treatment of AD has been considered a promising study area. There are limitations to metallic NP synthesis because of the chemistry involved, although metals including cerium, selenium, gold, and iron have been found to reveal significant anti-AD characteristics. Scientists are increasingly interested in the application of green chemical approaches in the design of NPs that are environmentally benign.

##### Selenium NPs

As previously stated, lowering the level of ROS in the brain is a significant strategy for treating AD. Various trace elements have been found as active ROS inhibitors that play a vital function in reducing oxidative stress and suppressing cell cytotoxicity because they are essential micronutrients in the human body and have a biological use in selenium nano formulation [185]. It has been discovered that reformed selenium NPs containing sialic acid can traverse the BBB and halt Aβ aggregation processes when exposed. Aβ aggregation can also be reduced by selenium nanoparticles modified by sialic acid and coated with peptide-B6 and epigallocate-3gallate (EGCG). It has been shown that a new nano formulation of selenium NPs encapsulated inside curcumin-enhanced PLGA nanospheres has strong anti-aggregation effects in a transgenic mouse model of AD [186]. 

##### Cerium NPs

Cerium oxide nanoparticles (CeONPs) can shield Alzheimer’s disease sufferers’ brains from reactive oxygen species overexposure (ROS). Using CeONPs in the treatment of AD has been found to have no negative effects and to be quite useful. In the therapy of AD, the use of cerium can be backed up by increased BBB uptake and no undesired buildup in other biological areas. Cerium NPs and triphenylphosphonium (TPP) prevent neuronal death in a preclinical AD animal model [187,188].

##### Gold NPs

Gold NPs are playing a vital role in delivering a drug through the BBB to the brain for the treatment of neurodegenerative diseases. In the AD mice model, D-glutathione-stabilized gold NPs can cross the BBB following intravenous administration as well as depict great inhibitory efficiency against Aβ42 aggregation, and also do not pose neurotoxicity [189]. Using intrahippocampal and intraperitoneal injections of gold NP formulation dramatically improves spatial learning and memory acquisition and retention [190]. The neuroprotective properties of dietary polyphenolic compounds like anthocyanin can be enhanced through the conjugation of anthocyanin with gold nanoparticles (NPs) [191]. 

##### Iron NPs

Biomedical research has extensively utilized iron oxide nanoparticles (NPs). It has been discovered that phenothiazine-based near-infrared (NIR) fluorescent dye combined with ultrasmall superparamagnetic iron oxide NPs can serve as a new theranostic agent for AD. By preventing Aβ plaque accumulation in the AD mouse brain, particles can be used to get NIR fluorescence and magnetic resonance imaging data [192]. The iron oxide NP formulations are also useful in the detection and treatment of neurological illnesses, such as AD [193]. 

NPs developed for the treatment of AD are summarized in Table 3 and various nanotechnological approaches are summarized in Figure 8.

##### Green Synthesized Nanoparticles

In the last five years, nanotechnologies have been approaching to the “green synthesis” of NPs, in line with the principles of green, cost-effective, and eco-friendly chemistry. This green synthesis approach replaces the use of toxic chemicals, as reducing and stabilizing agents, with phytochemicals during NPs’ synthesis [220]. Phytochemicals coating onto NPs’ surfaces guarantee their biocompatibility and bacteriostatic properties. Very recent studies demonstrated that green zinc oxide NPs maintained their high antioxidant properties; procyanidin fractions from *Leucosidea sericea* were employed in the synthesis of stable and active AuNPs, quercetin was used in the synthesis of gold and silver NPs, and, finally, silver NPs synthesized by macerating in the powdered fresh aerial parts of two plants, *Lampranthus coccineus* and *Malephora luteag*, demonstrated, in vitro, an anti-AchE and antioxidant activities comparable to rivastigmine, suggesting their application in AD [221].

### 5.5. Gene Therapy

AD gene therapy has gained a lot of attention recently. An enzyme or growth factor expressing gene was introduced as a generic treatment. Long-term expression of selected genes at therapeutic levels is a primary goal of this approach.

Neuroprotection and neuro-restoration can be achieved by altering or activating particular proteins that are implicated in the pathological process of neurodegenerative sickness. When it comes to treating neurodegenerative illnesses, gene therapy is an extremely complex operation that involves several variables, including time- and location-specificity, gene control, gene delivery, and more [222].

The manner of gene transfer (integrating vs. nonintegrating) and in vivo vs. ex vivo therapy varies depending on the target disease (genetic disease vs. complex acquired ailment). It is mostly accomplished through augmentation of the gene, inhibition of gene, and genome editing [223]. 

Further, there are tiny, single-stranded antisense oligonucleotides that link with RNA messengers to block the translation of a specific gene (also known as AS-Ons). It has been known for some time now that the antisense oligonucleotides known as IONIS MAPTRx, which are designed to limit the formation of tau, are antisense oligonucleotides that have been in clinical trials with the hope of serving as a unique way to lower tau production in the brain [224]. Antisense medication distribution to the brain is also a major difficulty in the treatment of AD.

#### Mechanism of Antisense Drug Delivery to the Brain

Amyloidosis in AD can be prevented with peptide nucleic acids (PNAs), which have a polypeptide backbone and can hybridize with target genes to limit amyloid formation and aggregation. Nonetheless, because PNAs are hydrophilic, they are unable to pass the blood–brain barrier. As a result, the receptor-mediated endocytosis process involving transferrin receptors on the BBB is a good fit for a method of carrying PNAs into the brain [225].

RNA interference (RNAi) is a technique used to interact with antibodies, such as anti-transferrin (OX 26) receptor monoclonal antibodies, which are then identified by specific transferrin receptors on the BBB. PNAs are an example of an antisense chemical. This method, which does not require the use of a transfection agent, promotes the uptake of PNAs into specific brain locations [226]. Recent research has demonstrated that fluorescent-labeled As-Ons can be taken up by dorsal root ganglionic neurons in the central nervous system (CNS) after an intrathecal injection without the use of a transfection agent (transfection agent). However, in comparison to cases where transfection agents were used, cellular uptake levels were shown to be lower [227,228].

Ataxin 2 is an RNA-binding protein found in RNA-containing granules, including stress granules, P bodies, and neuronal RNP granules. Ataxin 2 has a polyglutamine tract encoded by up to 23 cytosine-adenine-guanine (CAG) repeats. CAG repeat lengths of 34 or larger are associated with the severe neurodegenerative disease spinocerebellar ataxia 2 (SCA2). Intermediate-size expansions between 27 and 32 CAG repeats are associated with an increased risk of ALS. Decreasing the expression of *ataxin 2* using genetic methods or ASOs reduced neurodegeneration introduced in yeast, flies, and mice by increased expression of mutated *TDP-43*. As TDP-43 protein mislocalization to cytoplasmic aggregates is a common pathological feature for both sporadic and familial forms of ALS, reduction of *ataxin 2* by ASOs may be broadly beneficial as a therapy for most forms of ALS. In addition, SCA2 patients should also benefit from an *ataxin 2* antisense drug. Based on these findings, efforts are underway to identify and develop an antisense drug targeting *ataxin 2* for ALS and SCA2 [229,230,231,232]. 

As a result, establishing delivery systems that can enable cellular uptake followed by the continual release of these molecules for sustained activity is a major problem in such nucleic acid-based approaches. The FDA has approved nusinersen, an antisense oligonucleotide drug for SMD [229]. More research is needed in this area to produce compounds for AD as well. Carbon nanotubes are currently being used as a new technique for targeted medicine delivery. Additionally, drug delivery system (DDS) based on CN has been utilized to transport medications such as acetylcholine, as described below [233].

## 6. Conclusions and Future Prospective

In this comprehensive article, we have tried to gain an insight into the compounds targeting Aβ and tau proteins and neural pathways. Several treatments are at different stages of clinical trials, which offer certain prospects for AD treatment. The review intended to provide recent updates on therapeutic strategies in collected form for the management of AD, counting the use of medicinal plants, nutraceuticals, pharmacological approaches, nano pharmaceuticals, and gene therapy. Various remedies for neurodegenerative diseases such as AD have been known for a long time.

Currently, mechanistic studies are being conducted to authenticate and encourage the use of traditional medicines in animal models, and the majority of the herbs discussed in our review have been shown to have neuroprotective, antioxidant, and anticholinesterase actions. Nevertheless, most of the plant herbs are yet to be isolated, and more studies of these phytoconstituents have yet to be conducted, which is an intriguing feature in the treatment of AD.

Despite recent advancements in gene therapy and nanotechnology for the treatment of AD, the main challenge is the cost of these therapies which can be a barrier to accepting these recent techniques. To date, all available drugs are used for symptomatic treatment of disease. Hence, there is a critical call for the development of new drugs with new targets that can also prevent the disease progression at an early stage, leading to a better life for Alzheimer’s disease patients.

This review will certainly provide the researchers with a thorough understanding of AD, which will further help in designing compounds with a strong impact in curing AD.

## Figures and Tables

**Figure 1 molecules-27-04839-f001:**
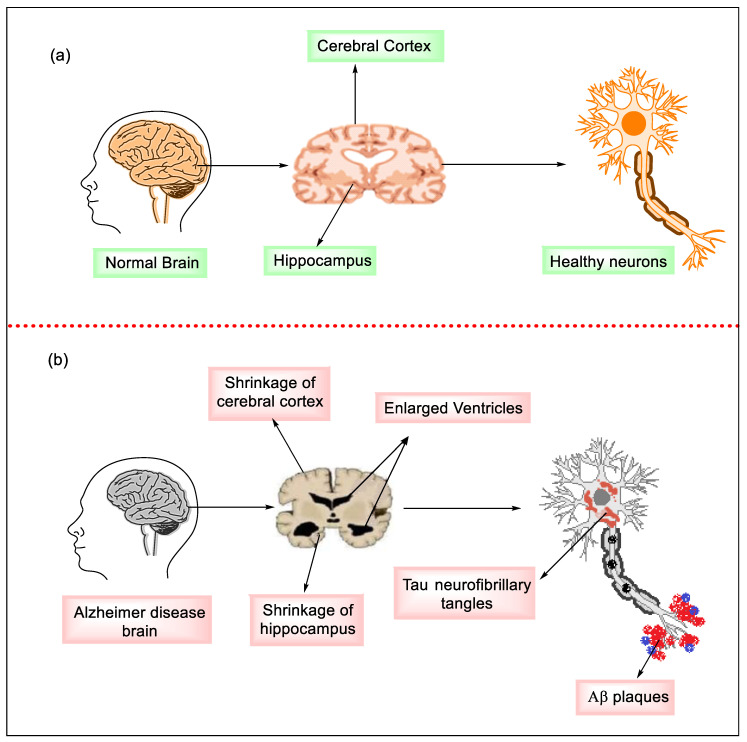
The physiological structure of the brain and neurons in (**a**) healthy brain and (**b**) Alzheimer’s disease (AD) brain.

**Figure 2 molecules-27-04839-f002:**
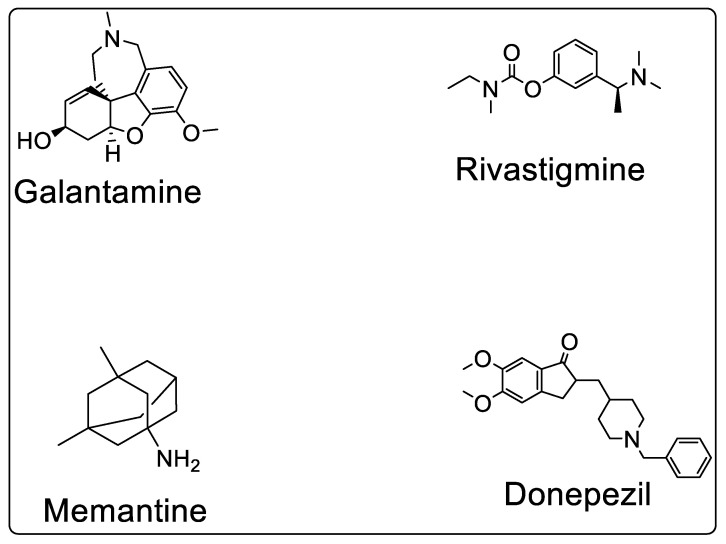
Chemical structures of FDA-approved anti-AD drugs.

**Figure 3 molecules-27-04839-f003:**
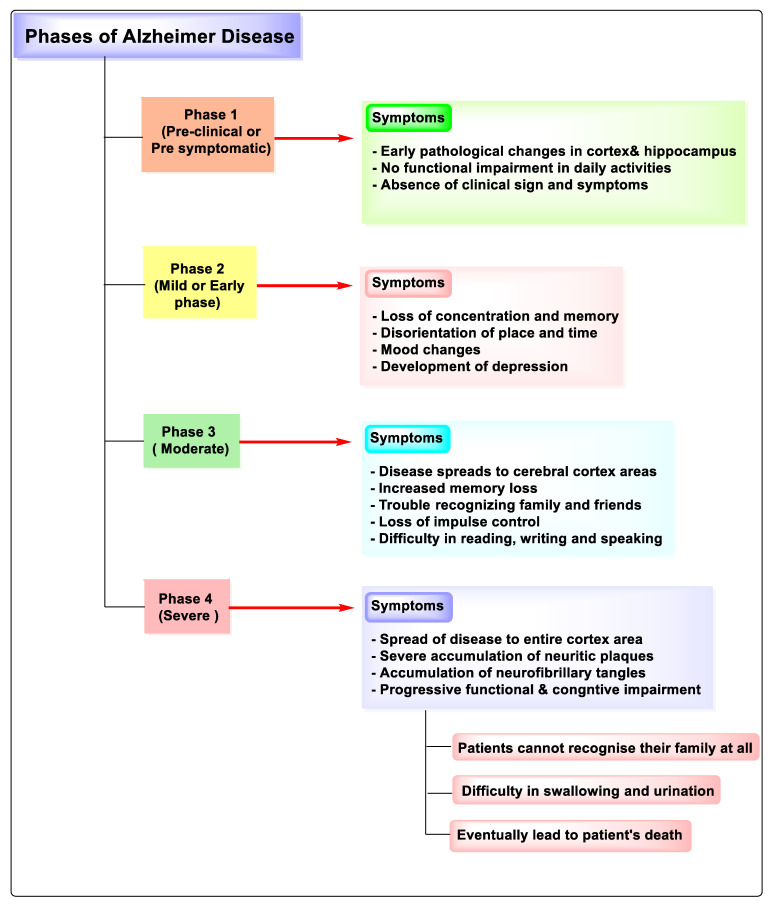
Phases of Alzheimer’s disease.

**Figure 4 molecules-27-04839-f004:**
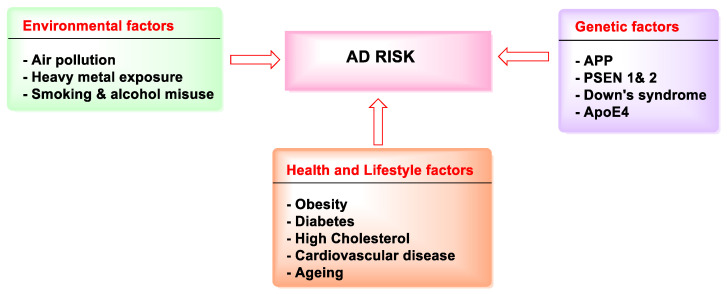
Pl Factors influencing the risk of AD development.

**Figure 5 molecules-27-04839-f005:**
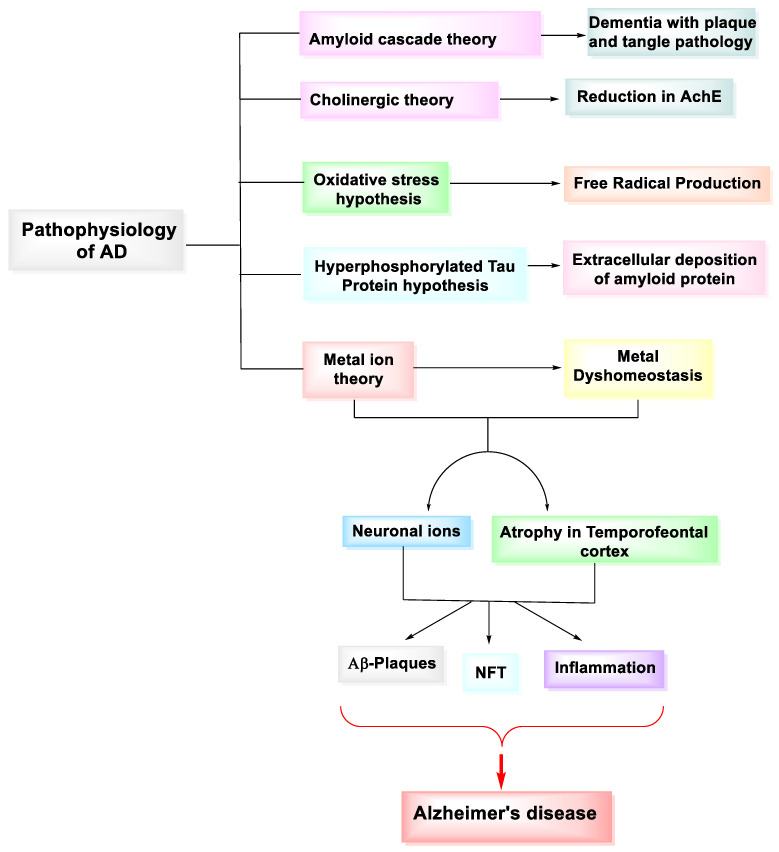
Hypothesis/Theories for the pathophysiology of Alzheimer’s disease.

**Figure 6 molecules-27-04839-f006:**
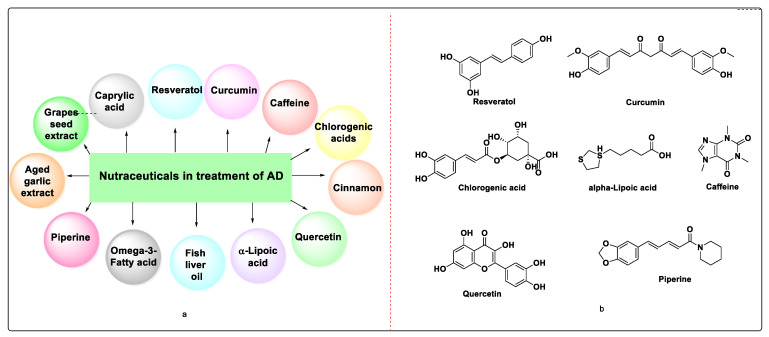
(**a**) Nutraceuticals used in the treatment of AD; (**b**) Structures of Nutraceuticals.

**Figure 7 molecules-27-04839-f007:**
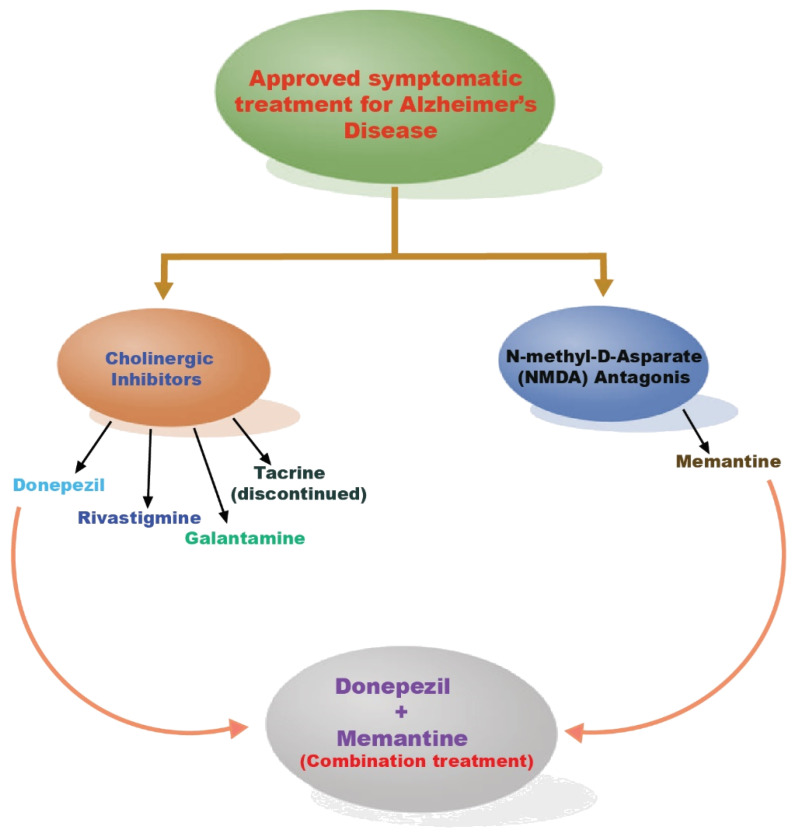
AChE inhibitors for the treatment of AD along with NDMA antagonist memantine.

**Figure 8 molecules-27-04839-f008:**
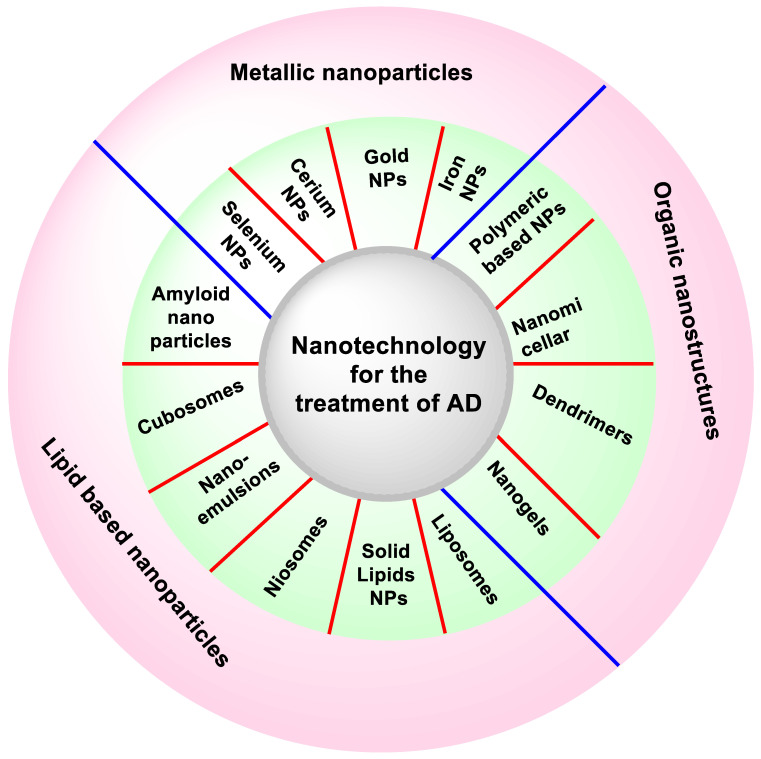
Different nanotechnology approaches for the treatment of AD.

**Table 1 molecules-27-04839-t001:** Some of the current clinical trials.

Investigational Drug/Nutraceutical (Category)	Mechanism of Action	Sponsor (Clinical Trial End Date)
Aducanumab (monoclonal antibody (MA))	Remove amyloid plaque	Biogen (April 2022)
Crenezumab (MA)		Roche/Genentech (July 2021)
Gantenerumab (MA)		Roche (November 2019)
Solanezumab (MA)		Eli Lilly (July 2022)
CNP520 (amyloid vaccine)	Reduces amyloid formation by inhibiting APP site cleavage	Axsome therapeutics (July 2024)
Methylphenidate (neurotransmitter-based)	Improves clinical symptoms by inhibiting dopamine re-uptake	Johns Hopkins (August 2020)
ABBV-8E12 (MA)	Removes tau protein	Abbvie (June 2021)
Cilostazol (vasodilator)	PDE3 antagonist	National Cerebral & Cardiovascular Center, Japan (December 2020)
Telmisartan (angiotensin receptor blocker)	Improve vascular functioning	Sunnybrook Health Sciences Centre (March 2021)
Deferipirone (iron chelator)	Reduces reactive oxygen species (ROS) that can harm neurons, making it neuroprotective	Neuroscience trials, Australia (December 2021)
Dronabinol (CB1 and CB2 endocannabinoids partial agonist)	Improve agitation (neuropsychiatric symptoms in AD)	John Hopkins University (December 2020)
Icosapent ethyl (purified form of Omega 3 fatty acid EPA (Omega 3 FA)	Neuroprotective, affords protection from disease pathology	University of Wisconsin (November 2021)
Grape seed extract (nutraceutical)	Anti-oligomerization agent	Mount Sinai AD Research Center (September 2020)

**Table 2 molecules-27-04839-t002:** Various herbs with their family, plant extract, phytoconstituents, and probable mechanism of action for the treatment of AD.

Botanical Name (Common Name)	Family	Plant Part/Extract Used	Phytoconstituent/s for Treatment of AD	Model/Method	Probable Mechanism of Action	Reference(s)
***Acorus calamus*** (Sweet flag)	Araceae	Roots; hydro-ethanolic (70:30) extract; methanolic Extract [70]	α and β asarone	Lipopolysaccharide-induced neuroinflammation in a model of rat [71]; Ellman method [70]	Reduces oxidative stress and has anti-inflammatory properties; inhibition of acetylcholinesterase (AchE)	[72,73]
***Allium sativum*** (Garlic)	Alliaceae	Bulbs	Di-allyl-disulfide and s-allyl cysteine	Scopolamine-induced amnesia; transgenic mouse model Tg2576	Anti-AchE activity, neuroprotective, antioxidant, hypocholesterolemic, reduces Aβ biomarker	[74,75,76]
***Angelica archangelica*** (Wild Celery)	Apiaceae	Methanolic extract of fruits	Furanocoumarins like imperatorin, xanthotoxin	Scopolamine-induced mice model [77]	Scopolamine-induced memory impairment is improved; the inhibition of AChE	[78]
***Bacopa monniera*** (Brahmi)	Scrophulariaceae	Hydroethanolic extract of leaves and stems; methanolic extract	Bitulinic acid, stigmasterol bacosides A & B, sapogenins & apigenin [79,80]	In vitro oxidative stress caused by aluminum in the hippocampus; oxidative damage caused by 3-nitropropionic acid	Inhibition of AchE improves cognition; enhances free radical scavenging mechanism; neuroprotective.	[81,82,83,84,85]
***Boswellia serrata***: (Salai guggul)	Burseraceae	Aqueous extract of Gum resin	Boswellic acids	Neurotoxicity caused by aluminum chloride in a rat model	Increase in acetylcholine levels in the brain	[86]
***Celastrus paniculatus*** (Black oil plant)	Celastraceae	Methanolic extract of Seed	Wifornine F, paniculatine A and B	AchE assay; DPPH antioxidant assay	Inhibition of AchE and DPPH assay signifies scavenging of free radicals	[87]
***Centella asiatica*** (Gotu kola)	Apiaceae	Aqueous extract of whole aerial plant	Asiatic and madecassic acid	STZ-induced oxidative stress and rat model of cognitive impairment	Lower level of Aβ biomarker which improves cognition and enhances antioxidant defense mechanisms	[69]
***Commiphora whighitti*** (Guggul)	Burseraceae	Ethyl acetate extract of resin	Guggulipid, guggulsterone	STZ-induced memory deficits model	Anti-AchE, anti-oxidant, and hypolipidemic activities	[88]
***Convolvulus pluricaulis*** (Shankhpushpi)	Convolvulaceae	Aqueous extract of roots	Shankhapushpine	Aluminum-induced neurotoxicity model; tau-induced neurotoxicity in the Drosophila fly model	Neuroprotection; a decrease in τ protein levels, i.e., a reduction in τ-induced oxidative stress is beneficial	[71]
***Curcuma longa*** (Turmeric)	Zingiberaceae	Methanolic extract of rhizome	Curcumin	Transgenic mice (chronic model) LPS-induced neuroinflammation model	Minimize interleukin-1 β levels, decreased Janus Kinase mediate transcription, and prevented Aβ aggregation Reduction in neuroinflammation due to anti-inflammatory properties	[70,77]
***Evolvulus alsinoides*** (Nela kuriji)	Convolvulaceae	Ethyl acetate extract of aerial part	Scopoletin	Shuttle box avoidance and step down paradigm and model	Nootropic activity	[89]
***Foeniculum vulgare*** (Fennel)	Apiaceae	Methanolic extract from fruits	Quercetin, rosmarinic acid, 3-caffeoylquinic acid, gallic acid, and kaempferol	Scopolamine memory deficit model	Increase in AchE inhibition	[90]
***Galanthus nivalis*** (Snowdrop)	Amaryllidaceae	Aqueous extract of bulbs	Galanthamine	Scopolamine-induced model [91]	AchE inhibitor activity	[92]
***Ginkgo biloba*** (Maiden hair tree)	Ginkgoaceae	Leaves	Ginkgoflavonglycosides & isorhamnetin	Amyloid precursor protein-transgenic mouse model	Neuroprotection by lowering APP levels	[93]
***Glycyrrhiza glabra*** (Licorice)	Fabaceae	Acetone extract of root	Glabridin	Scopolamine-induced model	Reduction in the brain cholinesterase activity	[94]
***Huperzia serrata*** (Toothed clubmoss)	Lycopodiaceae	Hydroethanol extract of aerial parts	Huperzine A	In albino male mice, there is memory loss	AchE inhibition, neuroprotection	[79]
***Lipidium Meyenii*** ***Walp*** (Peruvian ginseng)	Brassicaceae	Aqueous and hydroalcoholic extracts of hypocotyls	(1R,3S)-1-methyltetrahydro-beta-carboline-3-carboxylic acid	Scopolamine model	Inhibition of AchE activity	[80]
***Magnolia officinalis*** (Houpu magnolia)	Magnoliaceae	Ethanolic extract of bark	4-O-methylhonokiol	Aβ mouse model for neuronal toxicity	Neuroprotection	[95]
***Melissa officinalis*** (Common balm)	Lamiaceae	Ethanolic extract of leaves	Rosmarinic acid	Scopolamine-induced rat model	Inhibition of AChE activity	[96]
*Moringa olifera* (Drumstick tree)	Moringaceace	Hydro-methanolic leaf extract (20:80)	-	Hyperhomocysteinemia (HHcy) induced AD	Improved the homocysteine-induced oxidative stress	[97]
***Nardostachys jatamansi*** (Spikenard)	Valerianaceae	Ethanolic extract of roots	Nardostachysin, jatamansin, jatamols A and B	Diazepam and scopolamine induced amnesia in mice models	Increases cholinergic transmission, as well as neuroprotection and anti-oxidant activity	[98]
***Panax Ginseng*** (Ginseng)	Araliaceae	Alcoholic extract of roots	Gintonin, ginsenosides	Mice model for Aβ induced neurotoxicity	Activates the cholinergic system while inhibiting β and γ-secretase activity; reduced synaptophysin and choline acetyltransferase activity are restored to normal levels	[99]
***Phyllanthus acidus*** (Malay gooseberry)	Phyllanthaceae	Methanolic extract	-	Scopolamine-induced animal model of dementia and oxidative stress and elevated plus maze test	Decreasing lipid peroxidation and acetylcholinesterase activity	[100]
***Rosmarinus officinalis*** (Rosemary)	Lamiaceae	Hydroethanolic extract from leaves	Rosmarinic acid	Scopolamine-induced rat model	Decrease of AchE activity in the brain	[101]
***Salvia officinalis*** (Sage)	Lamiaceae	Methanolic extract of dry aerial parts	Carnosol, methoxyrosmanol, epirosmanol	Ellman method	AChE inhibition is dose-dependent	[102]
***Santalum album*** (Indian sandalwood)	Santalaceae	Oil	Alpha-santalol	TLC-bioautographic assessment and colorimetric method	Tyrosinase and cholinesterase inhibition	[103]
***Tinospora cordifolia*** (Giloya)	Menispermaceae	Aqueous extract of the whole plant	Choline	In a rat model, cyclosporine caused a memory loss	Immunostimulation as well as enhancing the synthesis of the neurotransmitter acetylcholine	[104]
***Urtica dioica* L.** (Stinging nettle)	Urticaceae	Hydromethanolic extract of leaves	Kaempferol, isorhamnetin, chlorogenic acid, scopoletin	STZ-induced diabetic mice model	Modulates glucose homeostasis in the hippocampus	[105]
***Withania somnifera*** (Ashwagandha)	Solanaceae	Aqueous and methanolic root extract	Withanamides	Amyloid peptide induced memory deficit	Cognition-enhancing and memory-improving effects	[106]
***Zingiber officinalis*** (Ginger)	Zingiberaceae	Methanolic extract of rhizomes	Flavonoid & polyphenol	DPPH method & Ellman method	Inhibition of acetylcholinesterase (AchE)	[107]

**Table 3 molecules-27-04839-t003:** Nanoparticles (NPs) developed for the diagnosis and treatment of Alzheimer’s disease: Classified on various hypothesis. (**a**) Amyloid cascade hypothesis and metal chelation therapy, (**b**) Oxidative stress, (**c**) Cholinesterase inhibitor, (**d**) Amyloid cascade hypothesis.

(**a**)					
**S. No**	**Nanoparticle**	**Application**			
1	MPB-PE & PDP-PE NPs [194]	Copper Aβ aggregates are solubilized when conjugated with D-penicillamine			
2	(CMC)-Nano-N2PY [195]	Accompanied by the pyridinone MAEHP, which is capable of dissolving and solubilizing iron aggregates.			
3	Nanoparticles coated with polysorbate 80 (CNPS and ICNPS) [196]	MHP: iron removal, ApoE, and ApoA-I binding			
(**b**)					
**S. No**	**Nanoparticle**	**Application**			
4	Nanoparticles (PLGA) [197]	CQ10 (coenzyme)-enriched NP			
5	A PEG-coated poly caprolactone core [198]	Resveratrol-loaded nanoparticles diminish Aβ associated toxicity in cellular systems			
6	PEG-GSH conjugate NP [199]	GSH NP: reduces oxidative stress in cells			
7	PLGA NP [200]	Encapsulated superoxidase dismutase NP: protection against H_2_O_2_-induced oxidative stress			
8	Solid lipid NP [201]	Aβ42 prevents the production of intracellular ROS by ferulic acid-loaded NP			
(**c**)					
**S. No.**	**Nanoparticle**			**Application**	
1	Chitosan NP, PBCA NP [202]			Tacrine-loaded NP	
2	PBCA NP with polysorbate 80 coating [203]			Rivastagmine-loaded NP	
3	Composite polystyrene/butylcyanoacrylate [204]			Aβ carrier of anti-acetylcholine esterase inhibitor PE154-loaded	
(**d**)					
**Amyloid Cascade Hypothesis**			**Amyloid Cascade Hypothesis**		
**S. No.**	**Nanoparticle**	**Application**	**S. No.**	**Nanoparticle**	**Application**
1	Polystyrene core PBCA shell [205]	Thioflavin T-loaded NP	9	DSPE-PEG2000 nanomicelle [206]	Inhibit Aβ aggregation, diminish beta sheet formation
2	Chitosan NP [207]	Aβ rich fragment: improves humoral immunity and decreases Aβ burden	10	Pullulan nanogel [208]	NP regulates Aβ aggregate formation
3	Chitosan polymeric NPs [209]	Anti-amyloid antibody-loaded	11	KLVFF functionalized nanodevice: KLVFFGG]_4_ peptide nanosheet [210]	Potentiates its inhibitory effect on Aβ1-42 aggregation
4	Gold NP [211]	Real-time detection, dissolves aggregates and fibrils	12	Polyamidoamine (PAMAM) [212]	Contains sialic acid
5	Maghemite NP [213]	Rhodamine- and Congo-red-loaded NP: selective Aβ fibril labeling	13	G3 PAMAM dendrimer [214]	It improves its inhibitory action on Aβ1-28 aggregation at minimum doses
6	Conjugate polymer NP [215]	Direct interaction with Aβ: aggregation modulation	14	(NiPAM:BAM) NP [216]	Fibrillation of amyloid-β Retarded
7	Gold NP (13) [217]	NMDA functionalized: inhibits Aβ aggregation	15	Sulfonated, sulfated, and fluorinated PS NP [218]	Aβ oligomerization
8	NP liposome [219]	Planar curcumine: capture Aβ and reduces toxicity			

## Data Availability

Not applicable.

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
