# Peer review of "Advances on Therapeutic Strategies for Alzheimer’s Disease: From Medicinal Plant to Nanotechnology"

_molecules, 2022, doi:10.3390/molecules27154839_

Round 1
Reviewer 1 Report
The paper by Hassan et al. discussed therapeutic strategies for Alzheimer’s disease through botanicals. The authors presented an interesting study on different therapeutic strategies for Alzheimer’s disease and its advances. Most of the sections are written well and justify the concept. The figures and tables are relevant but there are various areas where significant improvement is warranted, like reorganizing some parts of the paper (particularly the introduction section) so that may provide benefits to the readers of the related field.
Some other comments are shown
Although the content of the paper is fine, however, there are several issues with the language and quality needs extensive English editing for language and grammar throughout the paper.
In the introduction section, the global burden and prevalence rate of Alzheimer’s disease should be incorporated. Many of the ideas presented in the introduction blend together, with little to no real separation into focused paragraphs, making it challenging to decide what is the most important information to take forward, not clear, and needs to be re-write for better clarity.
In general, the introduction lacks development and could use better or more detailed explanations of the topics mentioned, such as describing the impact of Amyloidosis (given that the concept was deemed important enough to mention, to begin with). Explain this section thoroughly with suitable studies (literature).
In the figure section, figure 7 (AChE inhibitors for the treatment of AD along with NDMA antagonist memantine), the diagram is not up to the mark and needs to improve the quality of the figure. Please, modify the diagram accordingly.
Page 16, last paragraph (Nanotechnology and Treatment of AD). The authors explain the use of nanomedicine in the treatment of AD. However, no study is provided to justify it.
It could be interesting if authors also incorporate green synthesized nanoparticles as a therapeutic strategy for AD in the nanotechnology section (Nanotechnology and Treatment of AD).
Page 22, the second last section on gene therapy, requires reorganization, it is possible that the inclusion of existing literature with the most promising treatments could be useful to complete this section.
Page 23, section 6.6.1. (Mechanism of Antisense drug delivery to the brain), The authors explain the existence of amyloidosis in AD. However, no example/study is provided of this mechanism.
In the conclusion and future perspective section, the last paragraph should be elaborated on and discuss how this review will help researchers who are working on AD.
As throughout the manuscript, uniformity and connection between the paragraphs are missing.
In the reference, recent references must be given (at least more papers must be cited that are less than 5 years old).
Overall, major revision is required.
Author Response
Please, see the attachment

Reviewer 2 Report
In this review manuscript, Hassan et al. set out to describe the pathophysiology, targets, and therapeutics of AD. The manuscript does a thorough job including a variety of therapeutics in development for AD, making for a nice update on the literature all in one place. The primary weakness is the quality of the writing, detailed below.
General comments:
The authors do a good job covering the epidemiology, risk factors, phases and symptoms, as well as the pathophysiology of AD. They also thoroughly introduce various therapeutics that have been used as well as those that are under development. The gaps in understanding of disease onset and progression, as well as the utility of certain therapeutics are mentioned.
Much of the important content is included, however, there is significant redundancy throughout the manuscript, with no new added information each time a concept in introduced.
The manuscript is well structured overall, but would benefit from the writing being tightened up, and such conciseness would permit additional information to be added, particularly as specific therapies are covered. Many of the therapeutics would benefit from an introduction, explaining what it is, why it is relevant to AD, what the history of the compound is, how it has been used to treat AD thus far, and specifically stating what has yet to be done.
The cited references are a bit out of date, with 79 of them (36%) at least 10 years old. While some older references are expected to provide historical context and cite original findings, an emphasis on more recent references (within the last 5 years) would make the manuscript more relevant and up to date. Furthermore, at least 19% of the cited references are themselves review articles. In a review such as this, one would expect there to be more reliance on primary research articles. There is not an excessive number of self-citations.
While the figures are of sufficient quality and size, many of them do not significantly enhance the reader’s understanding of the literature presented. Figure-specific comments are below. Table 1 is very useful and much appreciated.
Aside from recapping previous findings, and identifying gaps and potential future directions, the authors do not make any novel or insightful conclusions about the studies they reference, or the field in general.
Specific comments:
In the early part of the manuscript, the authors go back and forth between stating AD can be cured/stopped or only alleviated symptomatically. At the end in the conclusion, they state drugs are only helpful in alleviating symptoms, not halting disease progression (lines 685-686). The manuscript overall should adopt this final tone, rather than using statements like “its progression can be stopped” (line 17), for which they provide no evidence. In addition,
In Figure 2, mild memory loss is listed as a symptom of Phase 1, however, how this does not contribute to functional impairment in daily activities or clinical signs and symptoms is unclear. This distinction needs to be made more clearly in the corresponding text (lines 58-62).
In Figure 3, it seems as though it would be more appropriate to have the center box labeled “AD Risk” and have the arrows going from the various factors (environmental, genetic, health and lifestyle) towards the AD box, rather than as they currently appear going towards the factors.
In section 3 (Pathophysiology, lines 144-176) and the accompanying Figure 4, there is no mention of the amyloid cascade hypothesis, despite it being discussed later in the manuscript.
In lines 182-3, the authors state “The depletion of the neurotransmitter is the primary cause of AD.” Do they really mean neurotransmitter loss causes AD directly, or is responsible for AD symptoms?
Lines 194-200 discuss memantine and donepezil, noting they were ineffective in mild to severe illnesses and only provide short-term relief. It would be helpful to include a rationale/reason/speculation for why this might be the case and potential avenues for future therapeutics.
Lines 207-208 - Please clarify whether AChE actually causes neurodegeneration (neuron loss), or simply decreased synaptic transmission (loss of synaptic elements, electrophysiology, etc).
Please provide an explanation and appropriate references for how AChE modulates amyloid fibril toxicity (lines 211-212).
In the discussion of GSK3 (lines 232-242), miR-124-3p is not introduced and the connection with GSK is not clear. Please clarify.
The section on BACE1 (lines 244-255) would benefit from some reorganization to make it more clear and concise. Furthermore, a discussion of therapies that have been tried that target BACE1 would make it more inclusive.
In the Neutraceutical section (starting on line 274), please define the term “neutraceutical” and give some context for how they could be significant in terms of usefulness in treating AD. What is the history and their association with AD risk/progression? How would a therapeutic dose compare to what is typically ingested? Why would some be more/less appealing as supplements?
Line 359 mentions SAMP8 mice. Please clarify/explain what they model.
In line 386, do AChEI’s alleviate, as previously discussed, or “cure” the symptoms of AD as stated? These are significantly different and need to be clear.
What is CHT1 (line 389)? It was not defined, nor its significance/relevance explained.
In section 6.3.1.2, the authors discuss BuChE (lines 400-401). What is it? Why is it important? Who expresses it?
Line 422 mentions “side effects, particularly in terms of learning and memory.” What are these side effects? How significant are they?
Figure 7 could be more informative if the drugs were plotted along an AD stage axis, showing where/when each one (or combination) is effective. They could further be color-coded by mechanism (cholinergic inhibitor = blue, NMDA antagonist = red).
Lines 429-439 introduce nanomedicine. This section is poorly written, and the information being conveyed is lost in the grammatical errors.
What is SpBMP-9 in line 459? How does it work?
What is coenzyme Q10 (line 464)? What is its significance/relevance?
In the introduction to dendrimers (lines 475-476), the term dendrimers needs to be defined/explained, as well as what low-generation dendrimers are and why that was what was tested. The context of what these are and why they are important is missing.
Why are lipid-based nanoparticles extensively contributing to the development of theranostic agents as stated in lines 493-404?
The introduction to liposomes (lines 511-512) is excellent.
The section on niosomes (lines 523-536) needs a lot of explanation/clarification. What are niasomes? What is Artemisia absinthium-loaded noisy-lipid nanocarrier systems? What is the fallout? Is AD truly primarily caused by folate deficiency? This was not mentioned in the various hypotheses for AD earlier in the manuscript.
The organization/structure of Table 2 is confusing and makes it difficult to read/understand.
What is DDS (line 660)? This is never spelled out/defined/explained.
There are many statements missing appropriate citations, including the first sentence in section 2 (lines 112-113), the doubling of mortality rates (line 119-120), evidence for redox transition balance (lines 164-165), among many others.
There are numerous typographical errors throughout the manuscript, including “control the AD” (line 22), capitalized “O” in obesity (line 84), “has crossed doubled” (line 119), as well as grammatical errors such as “there are many hypotheses for AD are” (line 142), and “are used in pharmaceutical companies” (line 259), to name a few.
Round 2
Reviewer 1 Report
Authors have successfully addressed the concerns and have revised the manuscript accordingly. Revised version looks improved and I have no hesitation to recommend it for publication.
Reviewer 2 Report
In this revised review manuscript, Hassan et al. have made many of the requested changes, which is much appreciated. However, there are still some issues that need to be addressed prior to the manuscript being considered ready for publication.
General comments:
One of the most significant changes was the inclusion of relevant background to set up various therapeutics. This greatly enhances the readers’ ability to understand why those compounds are discussed and how they work. However, much of the added description is lacking appropriate references/citation (see specific examples below). The cited references are still predominantly (130 of 221, 59%) older (more than 5 years since publication).
Specific comments:
In the Introduction and Epidemiology sections, the rates of incidence and prevalence of AD are noted, however, there is no explanation to a novel reader what these are nor how they could be different. Defining each term and explaining how one could increase while the other decreases would enhance the understanding of the reader (particularly one new to the AD field and more likely to read this manuscript).
In line 172, what is meant by “appears to be altering” is unclear.
The section on the Cholinergic hypothesis (lines 175-184) has many of the necessary parts, but there is no explanation of what the hypothesis states. This in fact comes much later in the cholinergic inhibitor section (lines 431-435) and would be much more beneficial to appear when it is first introduced.
In line 197 states “the actual pathways and mechanisms that cause AD are still unknown…” After just explaining the role of mutations in the APP and PSEN genes, this seems contradictory. If sporadic late onset AD is what was intended, please make this distinction.
In the section on BACE1, there is no discussion on what the other substrates are and whether/how those may contribute to difficulty in developing effective BACE1-directed therapies. The fact there are other substrates is mentioned but warrants additional information.
The sentence about the differences in nutraceuticals (lines 317-319) doesn’t make sense. How do they differ? From what are they different?
The intended meaning of the sentence “However, the impacts of these preparations on disease development biochemical procedures at the molecular stage have yet to be determined” (lines 336-337) is unclear.
In line 464, the authors state AD influences NMDA development. Do they mean it the other way – NMDA function influences the development of AD? Please correct, or explain what is meant.
Line 469 states “Moderate to severe AD can only be treated with Memantine.” This seems to be contradictory to the section above where Rivastigmine was said to treat mild to severe AD (line 454). Please clarify.
References/citations are needed for the following sections:
· MiR-124-3p (lines 259-262)
· BACE1 (lines 271-279)
· Donepezil (lines 441-444)
· Rivastigmine (lines 449-453)
· Nanomedicine introduction (lines 478-496)
· Dendrimers (lines 551-554 and lines 561-562)
· Lipid-based nanoparticles (lines 573-577)
· Niosomes (lines 609-614)
· Green synthesized nanoparticles (lines 700-713)
· Gene therapy (lines 718-730)
